# Development and Internal Validation of a Risk Prediction Model to Identify Myeloma Based on Routine Blood Tests: A Case-Control Study

**DOI:** 10.3390/cancers15030975

**Published:** 2023-02-03

**Authors:** Lesley Smith, Jonathan Carmichael, Gordon Cook, Bethany Shinkins, Richard D. Neal

**Affiliations:** 1Leeds Diagnosis and Screening Unit, Leeds Institute of Health Sciences, University of Leeds, Leeds LS2 9JT, UK; 2Cancer Research UK Clinical Trials Unit, Leeds Institute of Clinical Trial Research, University of Leeds, Leeds LS2 9JT, UK; 3NIHR (Leeds) Medtech & In Vitro Diagnostics Cooperative, Leeds LS2 9JT, UK; 4Department of Health and Community Sciences, Faculty of Health and Life Sciences, University of Exeter, Exeter EX2 5DW, UK

**Keywords:** myeloma, blood tests, risk prediction, early detection, reflex testing, primary care

## Abstract

**Simple Summary:**

Due to the rarity and non-specific symptoms of myeloma, a key challenge in diagnosing myeloma in primary care is the clinician considering myeloma and initiating appropriate investigations. We developed an algorithm to identify those at high-risk of having undiagnosed myeloma based on results from routine blood tests taken for other reasons. We demonstrated that it is possible to combine signals and abnormalities in several routine blood test parameters to identify individuals at high-risk of having undiagnosed myeloma with high predictive accuracy. Identified high-risk individuals could then undergo further automatic testing in the laboratory (a reflex test) to test specifically for myeloma. We estimate the number of additional tests needed to diagnose one case at different model thresholds. Further work is needed to explore the full potential of such a strategy. If successful, automatic reflex testing would enable large-scale, low-cost case-finding for myeloma with a significant impact on the diagnosis of myeloma.

**Abstract:**

Myeloma is one of the hardest cancers to diagnose in primary care due to its rarity and non-specific symptoms. A rate-limiting step in diagnosing myeloma is the clinician considering myeloma and initiating appropriate investigations. We developed and internally validated a risk prediction model to identify those with a high risk of having undiagnosed myeloma based on results from routine blood tests taken for other reasons. A case-control study, based on 367 myeloma cases and 1488 age- and sex-matched controls, was used to develop a risk prediction model including results from 15 blood tests. The model had excellent discrimination (C-statistic 0.85 (95%CI 0.83, 0.89)) and good calibration (calibration slope 0.87 (95%CI 0.75, 0.90)). At a prevalence of 15 per 100,000 population and a probability threshold of 0.4, approximately 600 patients would need additional reflex testing to detect one case. We showed that it is possible to combine signals and abnormalities from several routine blood test parameters to identify individuals at high-risk of having undiagnosed myeloma who may benefit from additional reflex testing. Further work is needed to explore the full potential of such a strategy, including whether it is clinically useful and cost-effective and how to make it ethically acceptable.

## 1. Introduction

Multiple myeloma is a cancer of the plasma cells, with approximately 6000 new cases diagnosed annually in the UK [1]. Myeloma has been identified as one of the hardest cancers to diagnose in primary care [2]. It is a relatively rare cancer; on average, a GP will see one new case every 8–10 years. Patients present with a range of non-specific symptoms, including back pain, bone pain, fatigue, and blood test abnormalities, such as hypercalcemia, renal impairment, anaemia, and raised inflammatory markers, which are all common in an aging population and may be falsely attributed to other common benign conditions [3,4]. Many myeloma patients have complex diagnostic journeys. Around half have three or more pre-referral consultations, and around one-third are diagnosed through emergency presentation [5,6,7]. Although the presentation may be non-specific, once considered, the diagnostic pathway for myeloma is relatively straightforward. However, delays in diagnosis result in a higher burden of disease-related complications, such as lytic bone disease, pathological fractures, acute kidney injury, and severe infection, all of which have been shown to limit the quality of life, treatment tolerance, response rates, and survival [5,8]. One-year survival ranges from 96% for those diagnosed with stage 1 disease to 80% for those diagnosed with stage 3 [9]. Improving the early diagnosis of myeloma is an area of unmet clinical need, and alternative diagnostic strategies are needed [10].

A range of blood tests are used to aid the diagnostic workup of myeloma [11]. Data from primary care records show that of newly diagnosed myeloma cases, 83% had urea and electrolytes (U&E) tests, 80% full blood count (FBC), and 52% calcium tests, with 49% having all three blood tests prior to diagnosis [12]. Analysis based on the National Cancer Diagnosis Audit found that prior to diagnosis, 65% of myeloma patients had U&E tests, 73% FBC, 58% liver function tests, and 49% tests for inflammatory markers [13]. Importantly, alterations in routine blood parameters can be detected several years prior to myeloma diagnosis, before the onset of symptoms or disease complications. Haemoglobin values have been shown to start to decrease in myeloma patients up to three years prior to diagnosis [14]. Both plasma viscosity (PV) and erythrocyte sedimentation rate (ESR) have been shown to have discriminatory suitability in the diagnosis of myeloma, with ESR and PV values starting to increase approximately two years prior to diagnosis [15].

Clinical prediction models have combined symptom data and blood test results to identify patients at higher risk of developing myeloma [3,16]. Both these studies were based on large primary care populations in England. Shephard et al. identified sixteen features independently associated with myeloma, each with generally low risk; however, once some symptoms were combined with leucopenia or hypercalceamia, risk estimates were strongly associated with myeloma [3]. Koshiaris et al. developed a risk prediction model combining symptoms and blood test results, including age, gender, back, chest and rib pain, nosebleeds, haemoglobin, platelets, white cell count, and mean corpuscular volume. The model had an area under the curve of 0.84 (95%CI 0.81 to 0.87); this increased to 0.87 (95%CI 0.84 to 0.90) when calcium and inflammatory markers were added to the model predictors. These models depend upon patients reporting specific symptoms to their GP and the GP coding them appropriately in the patient records for inclusion in the development of the model. Many myeloma symptoms are non-specific, and a rate-limiting step in diagnosing myeloma is the clinician considering myeloma and initiating appropriate investigations. Algorithms based on minor abnormalities and subtle changes in routine blood parameters only (such as haemoglobin, liver function tests, and inflammatory markers that may be taken for other reasons) may be useful to identify individuals with a high risk of having undiagnosed myeloma independent of symptom presentation and to prompt GPs into considering myeloma as part of the differential diagnosis. Such algorithms could then be implemented in hospital laboratories to trigger automatic reflex testing of high-risk individuals possibly having undiagnosed myeloma. A reflex test is a laboratory test performed after initial test results are obtained, and pre-determined criteria are met. Blood samples are stored in laboratories for up to 72 h after initial testing. Reflex testing of these samples based upon such pre-determined risk prediction criteria may enable large-scale, low-cost case-finding for myeloma for those at greatest risk without the need for additional clinics and blood specimens to be collected from patients.

A further consideration for strategies to improve the early diagnosis of myeloma is the potential increase in the diagnosis of the precursor condition monoclonal gammopathy of undetermined significance (MGUS). It is often diagnosed incidentally via routine blood tests. MGUS is common, with a prevalence of 3–5% and a risk of progression to myeloma of approximately 1% per year [17]. Once diagnosed, regular monitoring of blood tests and clinical symptoms is recommended, often guided by MGUS risk-stratification strategies. This has cost and workload implications for the NHS, as well as for the mental health of these patients, the majority of whom will never require treatment for MGUS.

Prior to developing and implementing a laboratory reflex test for myeloma, it is essential to investigate whether an appropriate algorithm based on results from routine blood tests can be developed with appropriate accuracy. In this study, we develop and internally validate a risk prediction model to identify those with a high risk of having undiagnosed myeloma based on results from routine blood tests taken for other reasons.

## 2. Material and Methods

### 2.1. Study Population

A retrospective case-control study used de-identified data from Leeds Teaching Hospitals Trust (LTHT), including clinical, pathology, and demographic data. Cases were defined as patients diagnosed with multiple myeloma (ICD-10 C90) between 2012 and 2019 at LTHT.

Controls were selected from LTHT patients who had at least two FBCs within a two-year time period from 2012–2019. Patients diagnosed with any other haematological cancer up to 2021 were excluded. Initially, five controls were selected for each case matched on age, gender, and year. The date of the last blood test was used as the index date. Further restrictions were applied to the control group to remove groups of patients who were critically ill and, therefore, may potentially have extreme outlying values for blood test results. This involved removal of controls where cancer was recorded as a comorbidity and those whose blood tests were all requested from intensive or palliative care.

All patients with an MGUS diagnosis recorded on their medical record (ICD-10, D47.2) between 2012 and 2019 were also identified for inclusion in a further sensitivity analysis.

### 2.2. Blood Tests

The blood tests included components of FBC and biochemistry blood tests. All tests carried out between 2011 and the diagnosis/index date were included, so cases and controls had blood tests at least one year prior to the index date. The blood tests included those requested from primary care and those conducted in hospital.

The total number and median number of blood tests for each individual were calculated, including all measurements up to five years prior to the diagnosis/index date. This analysis assessed how frequently these tests were conducted and to help select the final list of candidate predictors (Appendix A).

Although repeat measurements of blood tests over time were available, the prediction model was developed based on cross-sectional data, so we retained only one blood test result for each patient. For cases, the date the patient started to undergo tests as part of the diagnostic workup for their myeloma was identified and defined as the date of either a paraprotein, serum electrophoreses, or immunofixation test within the period 2 years prior to diagnosis. Only blood tests prior to this date were included, and the test result closest to this was retained. For controls, the test results closest to the index date were included.

### 2.3. Sample Size and Events per Parameter

The sample size was calculated based on the recommendations from Riley et al. to calculate the minimum number of events per parameter [18]. The prevalence of myeloma in this case-control study (after further exclusion of controls) was 27%, assuming an AUC of 0.80 (based on similar work by Koshiaris et al. [16]) and a shrinkage factor of 0.9. Based on these inputs, a minimum of 9.8 events per parameter was estimated. We included 375 cases and were able to include up to 38 parameter predictors. Therefore, we included age and sex and 15 blood tests to allow us to include non-linear terms for continuous variables.

### 2.4. Candidate Predictors

Candidate predictors initially considered were components of FBC and biochemistry blood tests (shown in Appendix A) that are commonly conducted in primary care and known to be abnormal at the point of a myeloma diagnosis. These were refined for inclusion in the modelling stage based on existing literature, clinical opinion, and data availability. The final list of variables to take forward to the modelling stage were selected via discussion with the clinical team, including a GP (RN) and haematologists (GC and JC). We only included routine blood tests well recorded within the data extract (recorded for at least 50% of cases and controls). The final fifteen blood tests included were: albumin, alkaline phosphatase (ALP), alanine transaminase (ALT), basophils, calcium, creatinine, C-reactive protein (CRP), eosinophils, haemoglobin, lymphocytes, mean cell volume (MCV), monocytes, neutrophils, platelets, and white cell count (WCC) (Appendix A).

### 2.5. Model Development

A risk prediction model was developed using logistic regression. A full modelling approach was adopted where all predictors were included in the model regardless of their statistical significance. This method avoids issues with potential overfitting and predictor selection methods, which may be data-dependent. To account for potential non-linearity, continuous predictors were modelled using restricted cubic splines with three knot points placed at the recommended locations based on percentiles (10th, 50th, and 90th percentiles) [19].

### 2.6. Missing Data

Multiple imputations using chained equations were used to account for missing data assuming the data were missing at random [20]. The imputation model included all predictors and outcomes using predictive mean matching for continuous variables. Model convergence was assessed by examining convergence plots and comparing the distributions of observed and imputed values. Twenty imputed datasets were generated, and analyses were conducted on each imputation set and pooled using Rubin’s Rule [20].

### 2.7. Internal Validation

The model was internally validated to assess the model discrimination and calibration. Model discrimination, which measures the ability of the model to differentiate between cases and controls, was measured using the C-statistic. Model calibration, which measures the agreement between predicted and observed risks, was assessed using calibration plots and the calibration slope. The calibration plot was obtained by generating 10 groups based on the deciles of predicted probability, plotting the predicted risks against the observed risks, and fitting a weighted smoothing trend line (loess smoother) between model predictions and outcome. The calibration slope was calculated by fitting a logistic regression for the outcome with the model linear predictor as the only independent variable. The coefficient for the linear predictor is the calibration slope (the ideal calibration slope is 1). Internal validation using bootstrapping with multiple imputation was conducted using 1000 bootstrap samples drawn from each imputed data set and results pooled to obtain the optimism-adjusted estimates.

### 2.8. Diagnostic Accuracy Statistics

Model sensitivity and specificity were obtained for probability thresholds ranging from 0.1 to 0.5 (by intervals of 0.05). The prevalence of myeloma in a primary care population (where the predicted algorithm would be implemented) was estimated based on published studies and official national statistics [12,16,21]. We estimated the positive predictive values (PPV), negative predictive value (NPV), number of cancers detected and missed, the total number that would require additional reflex testing, and the number needed to reflex test to diagnose one case at each probability threshold.

### 2.9. Sensitivity Analysis and Additional Models

Several additional models based on further exclusion criteria were run as sensitivity analyses. The additional models are listed below.

Calcium excluded: The model was run again, excluding calcium as a predictor. Calcium is an important predictor of myeloma [11]; however, it was the blood test that was conducted least in both the case and control group and had the highest levels of missing data. In clinical practice, if a GP is requesting a calcium test, they may already be thinking of the possibility of cancer.

Over 60s only: The model was run again, restricting the age of cases and controls to include those over 60 years of age only. The median age at diagnosis for myeloma is 71 years, with over 80% of new cases diagnosed in those over 60 years of age [21]. In our study cohort, there were 66 cases and 348 controls aged <60 years who were excluded from this analysis.

MGUS as controls: The MGUS cohort (n = 392) were excluded from the main model development. Our main aim is identifying myeloma cases, and we wish to avoid inadvertently detecting a large number of MGUS cases. The MGUS cohort were included in the control group, and the model was developed and validated using this updated control group.

Further models based on combinations of the above criteria were also run.

All ages, MGUS as controls, calcium excluded as a predictor.Over 60 s only, MGUS as controls, all predictors included.Over 60 s only, MGUS as controls, and calcium excluded as a predictor.

Each of these models was internally validated, and the model’s discrimination and calibration were assessed by deriving the C-statistic and the C-slope. The diagnostic accuracy statistics were calculated for each model as described above.

### 2.10. Software

Analysis was carried out in R version 4.0.2 using the following packages: dplyr for data manipulation [22], Hmisc and rms to generate descriptive statistics and model fitting [19], mice and nanair to explore missing data patterns and perform multiple imputations [23,24], pROC for model discrimination and estimating diagnostic accuracy statistics [25], and psfmi for model validation with multiple imputations [26].

The study is reported in accordance with Transparent Reporting of a multivariable prediction model for Individual Prognosis Or Diagnosis (TRIPOD) guidance [27].

## 3. Results

There were 439 myeloma patients diagnosed between 2012 and 2019, and, after all exclusions, 367 cases and 1488 controls were included in the prediction modelling (Figure 1). The baseline characteristics of the cohort are shown in Table 1. The median age of the cases was 71 years (IQR 63, 79), and 58% were male. Appendix A show the characteristics of the MGUS cohort. Appendix A show the trends and distributions of blood test results for cases and controls.

Approximately one-third of included blood tests were from primary care (Appendix A). The percentage of missing data for candidate predictors was greatest for calcium (35%) and CRP (28%), with a higher percentage of missing controls than cases. Data were missing for between 5–6% for albumin, ALP, and ALT, and for all other tests, the percentage with missing data was <1% (Appendix A).

### 3.1. Predictor Variables

Associations between each predictor and the outcome from the final model are shown in Figure 2, and the model coefficients are in Appendix A.

### 3.2. Model Discrimination and Validation

The C-statistic directly calculated from the dataset was 0.87 (95%CI 0.84, 0.89). After internal validation via bootstrapping, the C-statistic was 0.85 (95%CI 0.83, 0.87). The calibration slope was 0.87 (95%CI 0.75, 0.90), suggesting some model overfitting. The calibration plot is shown in Figure 3, and there is some suggestion of underestimation of risk in the groups with the highest risk. Appendix A shows histograms of the predicted probabilities for cases and controls.

### 3.3. Sensitivity Analysis

Model fit statistics for the series of sensitivity analysis models are shown in Table 2. The C-statistic ranged from 0.81 (95%CI 0.79, 0.84) for the model, including the MGUS cohort as controls and excluding calcium as a predictor, to 0.86 (95%CI 0.84, 0.89) for the model based on the over 60s and including all blood tests. The C-slope ranged from 0.86 to 0.88.

### 3.4. Diagnostic Accuracy

The diagnostic accuracy of all models at a range of threshold probabilities are shown in Table 3. At a probability threshold of 0.2, the sensitivity was 77.1 (95%CI 72.8, 81.2), and the specificity was 82.3 (95%CI 80.3, 84.3); increasing the threshold to 0.4 resulted in the sensitivity reducing to 58.3 (95%CI 53.1, 63.5) while the specificity increased to 94.8 (93.6, 95.9).

Assuming a prevalence of 15 cases per 100,000 population and a probability threshold of 0.2, the number of additional reflex tests to be carried out ranges from 14,009 to 21,508 depending on the chosen model, with the corresponding number needing a reflex test to diagnose 1 case ranging from 1269 to 1853. Increasing the probability threshold to 0.4 decreases the total number of additional reflex tests to between 4006 and 5807, with the number needed to reflex test to diagnose 1 case ranging from 573 to 704. Appendix A show these metrics for different prevalence estimates.

## 4. Discussion

### 4.1. Key Findings

We have developed a risk prediction model to identify individuals at high-risk of having undiagnosed myeloma based on the results of blood tests taken for other reasons. The model shows promising performance after internal validation and sensitivity analysis with good discrimination and calibration. Individual symptoms and blood test abnormalities have low predictive value. However, in this proof-of-concept study, we showed it is possible to combine signals and abnormalities in several routine blood test parameters to identify individuals at high-risk of having undiagnosed myeloma who may benefit from additional reflex testing specifically for myeloma.

The model is based on blood tests commonly conducted in primary care. Based on the model parameters, the number of patients identified for additional reflex testing varies considerably. At a probability threshold of 0.1, approximately 40,000 individuals would be identified to undergo further testing; this reduces considerably to around 5000 individuals at a probability threshold of 0.4, with the corresponding number needed to diagnose one case reducing from around 3000 to 600. However, at higher probability thresholds, the number of cancers missed increases.

### 4.2. Comparison with Other Studies

Previous prediction models have focussed on including blood test results alongside presenting symptoms to identify high-risk patients. Our study was not reliant upon patients reporting specific symptoms, so it is not completely comparable. However, the previously developed models included several blood test abnormalities, including anaemia, leucopoenia, hypercalcemia, and raised inflammatory markers [3,16].

Raised inflammatory markers are associated with increased cancer risk [28]. Specifically for myeloma, previous research has shown that PV and ESR are more useful for diagnosis than CRP [15]. In our model, we included CRP rather than PV or ESR, as this was the most commonly used inflammatory marker in our cohort. PV was missing for 31% of cases and 73% of controls, and ESR was missing for 76% of cases and 87% of controls. These tests are also more likely to be performed in patients where cancer is suspected, and therefore, their predictive value may appear higher. While methods to deal with missing data could have been used to impute missing values and allow these markers to be included in the model development, we felt that since these tests were only conducted in a small proportion of the study sample when the model algorithm is implemented clinically, this would lead to many exclusions and therefore decided not to include these in the model development. This is also supported by data from CRPD which shows that in the prediction model developed by Koshiaris et al. ESR was missing for 74%, CRP was missing for 81%, and PV was missing for 96% of the study population [16], and Watson et al. showed that PV was the least commonly used inflammatory marker [28].

Hypercalcaemia plays an important role in the diagnosis of myeloma and is one of the CRAB features essential for the diagnosis of myeloma [29]. Out of all blood tests included in our model, this is the one conducted the least (84% in cases and 60% in controls). Patients with hypercalcaemia have an increased risk of cancer [30], and it may be that when this test is requested, the clinician is already thinking about the possibility of a cancer diagnosis. For these reasons, we re-ran our model, excluding calcium as one of the predictors, and we still obtained similar model discrimination and calibration. Again, this is important for implementing the developed model so that patients are not excluded if they do not have the included blood tests.

### 4.3. Clinical Implications

A major challenge in diagnosing myeloma in primary care is the clinician suspecting myeloma and appropriately testing for it. Our proposed approach based on reflex testing removes this step by identifying high-risk individuals on blood tests already being conducted with minimal impact on NHS resources. However, there are challenges and several issues to consider with this strategy. Reflex and reflective testing are common in UK laboratories, but there is variation between laboratories and different approaches used in decisions to contact the requestor before adding a particular test [31]. Whether this test would be considered automatically, based on our developed algorithms, or whether it is dependent upon the laboratory specialist adding the additional tests and the mechanisms for this need to be defined. Setting appropriate thresholds for the test is challenging, along with risk communication. There are serious ethical issues around the potential diagnosis of a disease that was not originally considered when the blood test was requested. Further work is needed to explore all these issues with a range of stakeholders, including patients and the wider population, to determine whether this strategy is clinically useful and cost-effective and how to make it ethically acceptable.

The number of individuals to undergo additional testing varies considerably depending on the probability threshold chosen. Further economic evaluation work is needed to explore whether this type of automatic triage testing pathway can be cost-effective and establish the necessary appropriate thresholds that incorporate the costs of the additional tests. This study provides the evidence base to conduct this health economic analysis which we are undertaking as the next stage of this project. One advantage of this method is that the additional test used in the diagnostic workup for myeloma, serum protein electrophoresis, immunofixation, and serum-free light chain assays are sensitive and specific initial tests for the majority of patients and relatively inexpensive compared to other diagnostic methods.

While trying to improve the early diagnosis of myeloma, it is important to avoid inadvertently diagnosing a large number of MGUS cases which would have a significant impact on patients’ mental health as the majority of patients will never require treatment for MGUS. It also has significant cost and workload consequences for the NHS. We included the group of patients with an MGUS diagnosis recorded in sensitivity analysis combined with the control group and found that the model had similar predictive performance to the model, excluding this group. It should also be acknowledged that the control group may include patients with undiagnosed MGUS. A population-based screening trial for MGUS is ongoing in Iceland, where participants with MGUS are randomised to different follow-up strategies, providing vital evidence on the cost-effectiveness of screening and monitoring approaches [32].

### 4.4. Strengths and Limitations

The key strengths of this study are the methods used in model development. All blood tests included were those commonly conducted in primary care and undertaken for reasons other than a cancer diagnosis; they were not reliant upon the patient reporting specific symptoms and the clinician considering myeloma. All test results were considered on a continuous scale, and we did not use reference ranges to define abnormal results, which may vary across laboratories. The dichotomisation or categorisation of data also results in a loss of statistical power [33]. Spline functions were used to model the relationship between predictors and the outcome rather than assuming a linear relationship [34], and multiple imputations were used to account for missing data [20].

The major limitation of our study was the selection of the control group. This was based on hospital patients who had at least two FBCs over two years. These patients may not be representative of the wider population that our developed model would be implemented into. The blood test data we had access to were not limited to those conducted in hospital only, and around one-third of those included were requested by GPs. We tried to minimise some of this bias further by excluding tests with extreme values indicative of critical illness. Furthermore, the planned reflex testing would only be applied to patients undergoing blood tests who may represent a sicker population, and there is evidence to suggest an increased cancer risk in patients who undergo testing even if the test results are within the normal range [28,35,36,37]. We adopted a case-control approach to ensure a sufficient number of cases for robust model development rather than a cohort study design. Further validation of our developed model in a primary care population is needed by adopting a cohort approach.

Full data was not available for all tests included in the model. While we overcame this in the model development using methods to deal with missing data, there may be implications for the implementation. If patients have not undergone the required blood tests, then we may not be able to calculate their predicted risk. Due to sample size limitations, we could only use up to 15 blood tests. Other tests may have been important, including those used to determine myeloma prognosis, but these tests were less likely to be conducted prior to diagnosis, again limiting the clinical utility of the developed model.

The model was based on cross-sectional data using results from one time point for each individual. For myeloma patients, we based this on those prior to the initiation of the diagnostic work for myeloma. Changes in some blood tests can be seen to 2–3 years before diagnosis [14,15]. Further work is required to identify whether such longitudinal trajectories and changes can be incorporated into a risk prediction model, with a growing interest in methods incorporating repeated measures captured within electronic health records [38,39].

## 5. Conclusions

To improve the timeliness of myeloma diagnosis, it is essential to identify high-risk individuals and initiate appropriate investigations. In this proof-of-concept study, we showed it is possible to combine signals and abnormalities in several routine blood test parameters to identify individuals at high-risk of having undiagnosed myeloma who may benefit from additional reflex testing. Further work is needed to explore the full potential of such a strategy. If successful, automatic reflex testing would enable large-scale, low-cost case-finding for myeloma, with a significant impact on myeloma diagnosis.

## Figures and Tables

**Figure 1 cancers-15-00975-f001:**
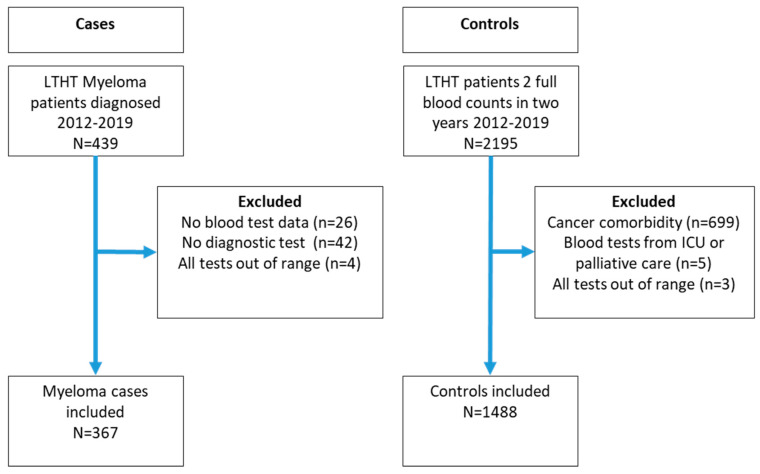
Flow chart of study participants.

**Figure 2 cancers-15-00975-f002:**
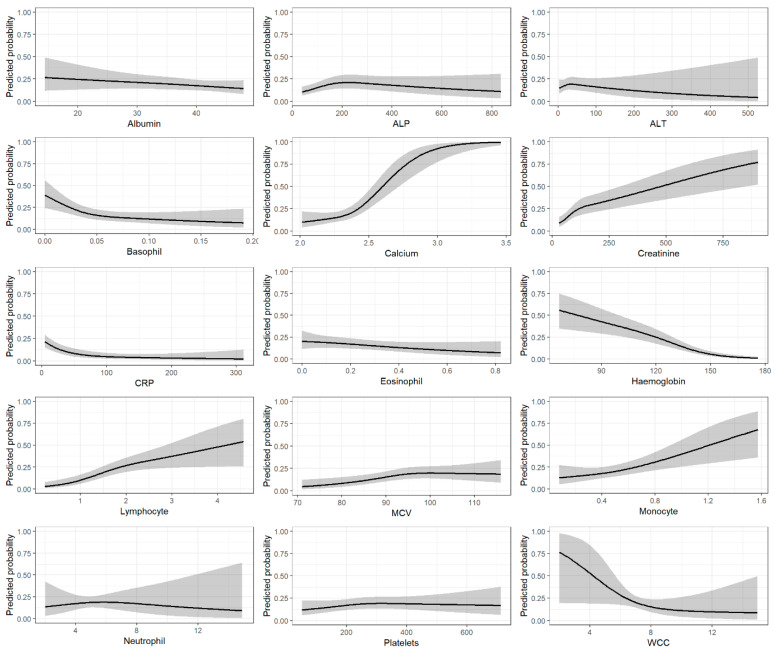
Multivariable associations between blood tests and predicted probability of myeloma.

**Figure 3 cancers-15-00975-f003:**
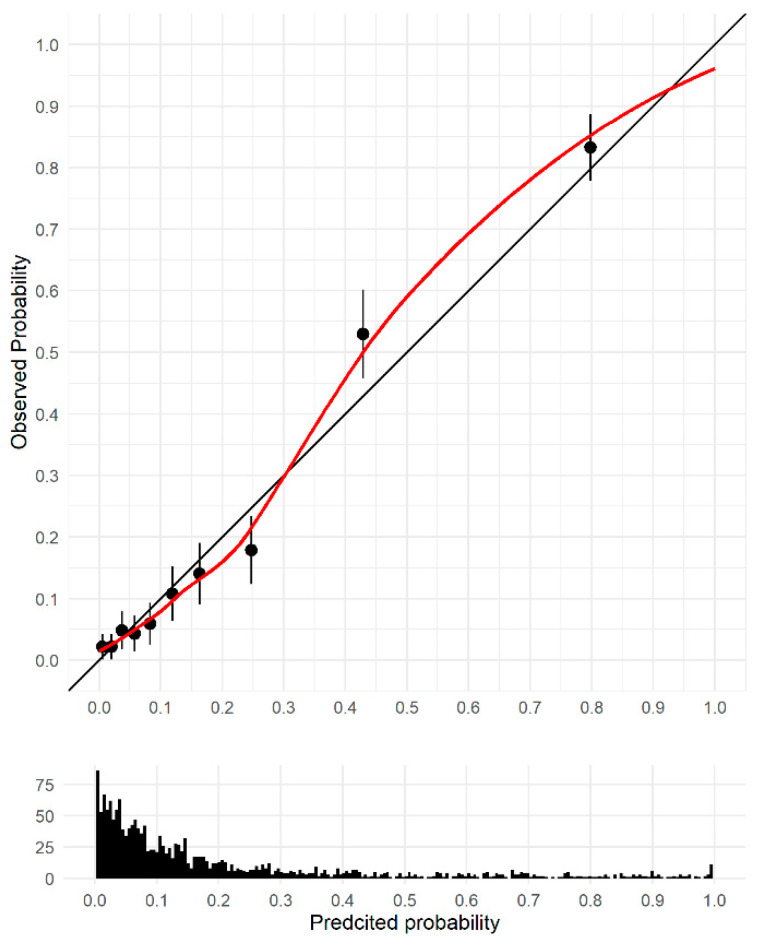
Calibration plot, pooled over multiple imputed datasets. Circles represent deciles of patients grouped by simialr predicted risk, the red line shows the loess smoothed line and the distribution of the predicted probabilty for all patients is shown in the histogram at the bottom of the graph.

**Table 1 cancers-15-00975-t001:** Study cohort description.

Variable	Cases N = 367		Controls N = 1488	
Age, median (IQR)	71	(63, 79)	70	(60, 78)
Age group, n(%)				
<50 years	18	4.9%	111	7.5%
50–59	48	13.1%	237	15.9%
60–69	98	26.7%	389	26.1%
70–79	125	34.1%	452	30.4%
80–89	71	19.3%	275	18.5%
90+	7	1.9%	24	1.6%
Sex, n(%)				
Male	214	58.3%	875	58.8%
Female	153	41.7%	613	41.2%
Ethnicity, n(%)				
White British	309	84.2%	1229	82.6%
Black	17	4.6%	15	1.1%
South Asian	10	2.7%	68	4.6%
Other	16	4.4%	30	2.2%
Missing	15	4.1%	146	9.8%
Total number of comorbidities, n(%)				
0	242	65.9%	1018	68.4%
1	88	24.0%	326	21.9%
2	29	7.9%	111	7.5%
3+	8	2.2%	33	2.1%
Blood tests, median (IQR)				
Albumin	36.0	(31.0, 40.0)	39.0	(34.0, 42.0)
ALP	139.0	(89.0, 201.0)	138.0	(83.0, 213.0)
ALT	19.0	(14.0, 30.0)	21.0	(15.0, 31.0)
Basophils	0.03	(0.02, 0.04)	0.04	(0.03, 0.06)
Calcium	2.41	(2.31, 2.54)	2.33	(2.27, 2.40)
Creatinine	89.50	(72.0, 143.2)	76.00	(63.0, 95.25)
CRP	5.00	(5.0, 19.62)	15.10	(5.00, 54.50)
Eosinophils	0.12	(0.06, 0.19)	0.13	(0.06, 0.23)
Haemoglobin	112.0	(97.0, 127.2)	132.0	(115.0, 144.0)
Lymphocytes	1.49	(1.07, 2.03)	1.46	(1.03, 1.99)
MCV	96	(92, 101)	94	(90, 98)
Monocytes	0.39	(0.29, 0.50)	0.45	(0.34, 0.57)
Neutrophils	3.64	(2.75, 5.28)	4.95	(3.69, 6.94)
Platelets	223.0	(167.0, 288.5)	248.0	(199.0, 309.0)
WCC	6.14	(4.73, 7.94)	7.58	(5.95, 9.73)

Abbreviations: IQR = Interquartile range, ALP = alkaline phosphatase, ALT = alanine transaminase, CRP = C-reactive protein, MCV = mean cell volume, WCC = white cell count.

**Table 2 cancers-15-00975-t002:** Model discrimination and calibration for the full model and all sensitivity analyses.

Model	C-Statistic	C-Slope
Full model (all 15 blood tests included)	0.85 (0.83, 0.87)	0.87 (0.75, 0.90)
Calcium removed as predictor	0.83 (0.81, 0.85)	0.86 (0.73, 1.01)
Over 60s only included	0.86 (0.84, 0.89)	0.86 (0.73, 1.01)
MGUS cohort included as controls	0.83 (0.81, 0.86)	0.88 (0.76, 1.00)
All ages, MGUS as controls, calcium removed	0.81 (0.79, 0.84)	0.88 (0.75, 0.99)
Over 60s only, MGUS as controls, all predictors	0.85 (0.82, 0.87)	0.87 (0.75, 1.00)
Over 60s only, MGUS as controls, calcium removed	0.83 (0.80, 0.85)	0.87 (0.73, 1.00)

**Table 3 cancers-15-00975-t003:** Diagnostic accuracy metrics for all models at a range of probability thresholds assuming a prevalence of 15 cases per 100,000 population.

				Prevalence 15 per 100,000					
Model	Threshold	Sensitivity (95% CI)	Specificity (95% CI)	PPV	NPV	Total to Reflex Test	Cancers Diagnosed	Cancers Missed	Number Needed to Reflex Test to Detect 1 Case
Full model	0.1	90.2 (86.9, 92.9)	60.2 (57.6, 62.8)	0.0003	1.0000	39,808	14	1	2942
	0.2	77.1 (72.8, 81.2)	82.3 (80.3, 84.3)	0.0007	1.0000	17,709	12	3	1531
	0.4	58.3 (53.1, 63.5)	94.8 (93.6, 95.9)	0.0017	0.9999	5208	9	6	596
Calcium removed	0.1	88.8 (85.8, 92.1)	55.6 (53.0, 58.0)	0.0003	1.0000	44,407	13	2	3334
	0.2	77.4 (73.3, 81.5)	78.5 (76.3, 80.5)	0.0006	1.0000	21,508	12	3	1853
	0.4	55.0 (50.1, 60.2)	94.2 (92.8, 95.2)	0.0014	0.9999	5807	8	7	704
Over 60s only	0.1	91.4 (88.0, 94.4)	61.6 (58.9, 64.2)	0.0004	1.0000	38,408	14	1	2801
	0.2	81.7 (77.1, 86.1)	80.6 (78.5, 82.9)	0.0006	1.0000	19,409	12	3	1584
	0.4	65.1 (59.5, 70.4)	94.3 (92.9, 95.6)	0.0017	0.9999	5709	10	5	585
MGUS as controls	0.1	84.2 (80.1, 87.7)	63.9 (61.7, 66.0)	0.0003	1.0000	36,107	13	2	2859
	0.2	73.6 (69.2, 77.9)	86.0 (84.3, 87.5)	0.0008	1.0000	14,009	11	4	1269
	0.4	46.6 (41.7, 51.8)	96.0 (95.1, 96.9)	0.0017	0.9999	4006	7	8	573
MGUS as controls and calcium removed	0.1	85.6 (82.0, 89.1)	59.4 (57.1, 61.6)	0.0003	1.0000	40,607	13	2	3163
	0.2	71.7 (67.0, 76.3)	82.8 (81.1, 84.5)	0.0006	0.9999	17,208	11	4	1600
	0.4	44.1 (39.5, 49.1)	95.7 (94.7, 96.6)	0.0015	0.9999	4306	7	8	651
Over 60s only, MGUS as controls, all predictors	0.1	87.0 (83.1, 90.7)	64.0 (61.5, 66.4)	0.0004	1.0000	36,008	13	2	2759
	0.2	75.4 (70.4, 80.0)	84.6 (82.8, 86.3)	0.0007	1.0000	15,409	11	4	1362
	0.4	48.8 (43.2, 54.5)	94.9 (93.7, 96.0)	0.0014	0.9999	5107	7	8	698
Over 60s only, MGUS as controls, calcium removed	0.1	87.0 (83.4, 90.7)	60.7 (58.1, 63.1)	0.0003	1.0000	39,307	13	2	3012
	0.2	74.8 (70.1, 79.7)	81.5 (79.4, 83.4)	0.0006	1.0000	18,508	11	4	1650
	0.4	48.5 (43.2, 54.2)	94.6 (93.5, 95.7)	0.0013	0.9999	5406	7	8	743

Abbreviations: CI = confidence interval, PPV = positive predictive value, NPV = negative predictive value.

## Data Availability

The data are not publically available due to restricted access. The data used for this research project are available to the research community via application to LTHT Research Data and Informatics team. The R code used in the analysis can be found at https://github.com/LFairleySmith/MyelomaPrediction (accessed on 25 January 2023).

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
