# Peer review of "Development and Internal Validation of a Risk Prediction Model to Identify Myeloma Based on Routine Blood Tests: A Case-Control Study"

_cancers, 2023, doi:10.3390/cancers15030975_

Round 1
Reviewer 1 Report
Smith et al propose a risk prediction model to identify patients at high risk of developing MM using 15 blood parameters that are commonly available in a general practitioner’s setting. They advocate that using this model, the performance of an additional reflex testing would allow to identify 1 MM patient among 600 tested individuals.
The issue is of interest and fits well in the scope of the special issue. I have some comments and concerns.
My main concern is the clinical utility of the proposed model. If I understood correctly, there is a detailed description of the statistical methodology employed and the prediction result and the 15 variables included are shown. However, I did not understood how to identify individuals at risk in the every day clinical setting.
Moreover, I also have some doubts about the cost-benefit ratio, given that it is necessary to test 600 individuals to identify 1 MM patient
Instead of high MM patients, it would probably be more adequate to say patients at high risk of developing MM.
Reviewer 2 Report
This manuscript, written by Dr Smith, original research, with the title of "Development and internal validation of a risk prediction model to identify myeloma based on routine blood tests: a case-control study" analyzed a large series of patients with multiple myeloma and controls to create a model to predict the presence of disease using commonly used variables. The manuscript is well written, it is easy to read, and to understand. Since the variables to be used are blood tests, there is potential use in the clinical routine by GPs.
As I understood, the authors used the C-statistic (also known as the "concordance" or C-index). This statistic is a mesure of "goodness of fit" for binary logistic regression models. In clinical studies, the c-statistic gives the probability. It is like the area under the curve and it ranges from 0.5 to 1. Over 0.7 is a good model. Over 0.8 is strong. A value of 1 is perfect prediction. As shown in Table 2, the c-statistic was above 0.8 in all models.
Comments:
1) In Table 1, could you please compare the variables between the cases and controls. Could you please calculate the p values and highlight the relevant variables?
2) If you calculate a binary logistic regression, backward conditional, using the same input variables (predictors), do you get similar results? You could use SPSS for this purpose.
3) How many missing data was present in the database? If you run the analysis excluding the missing data, how many cases remain, and does the result change?
4) In this model, among all the variables that were included in the final model, what were the most relevant ones? What should a GP used to predict the diagnosis of myeloma?
5) Did you try a decision tree?
6) Could you please describe the different abbreviations in the "footnote" of the different tables? For example, in Table 3, the CI, PPV, NPV.
7) Is it possible to explain the pathological relevance for myeloma pathogenesis of the different 15 blood test included in the final model?
Reviewer 3 Report
The authors develop a model and internal validation of a risk prediction model to identify myeloma based on routine blood tests.
Points to be addressed:
1) The rationale of why the authors came up with this research is scanty and is related to a lack of novelty: please highlight what this manuscript might add.
2) What is the information that is not exactly available that motivated the authors to come up with this information. What are the current caveats and how do the authors highlight the current research in answering them? If not they need to address in background and infuture directions .
3)State of the art figures are required: scale bar should be provided in high resolution.
4)The authors could provide a little more consideration of genomic directed stratifications in clinical trial design and enrolments.
5)The underlying message here is that more precision and individualized approaches need to be tested in well-designed clinical trials – a challenge, but I would be interested in their perspective of how this might be done. If beyond the scope of the manuscript, this should be highlighted as a limitation
6) The authors need to highlight what new information the review is providing to enhance the research in progress
7) how their risk impact on clinical practice compared to ISS, R-ISS, R2-ISS
8) do the author envision a role of their risk stratificator referring to high risk myeloma intended by bone morrow emancipated phenotypes (i.e. extramedullary disease?)
9) This reviewer, in the frame of point 8 thinking, personally misses some insights regarding the role of the MM microenvironment in potentially impacting the authors findings: Complex interactions between tumor plasma cells and the bone marrow (BM) microenvironment contribute to generate an immunosuppressive milieu characterized by high concentration of immunosuppressive factors, loss of effective antigen presentation, effector cell dysfunction, and expansion of immunosuppressive cell populations, such as myeloid-derived suppressor cells, regulatory T cells and T cells expressing checkpoint molecules such as programmed cell death. Considering the great immunosuppressive impact of BM myeloma microenvironment, many strategies to overcome it and restore myeloma immunosurveillance have been elaborated.
Please refer to PMID: 33194767 and expand the introduction/discussion sections
Reviewer 4 Report
Abstract:
- Please combine the summary and the abstract together. The summary is not necessary, it reiterates similar information to the abstract.
Introduction:
- Please provide a table, summarizing the algorithms used by contemporary authors, data size, and the results.
Which data imputation method was used?
Figure 2 is out of range. Please fix it.
Are the data balanced for each the classes? If not did the authors use any method to do so?
Results:
- Please provide a validation curve, to show that no overfitting and underfitting issues exits because of the data imbalane.
- In table 3 please indicate clearly the accuracy, F1 score, AUC.
- Are the results training accuracies or cross validated results. Please provide cross-validated results and mention which K-fold validation is implemented.
References:
- Some articles are more than 7 years old. Please use more recent ones.
Round 2
Reviewer 1 Report
No additional ones to those raised in the previous revision